# Robustness Evaluation of Multi-Agent Reinforcement Learning Algorithms using GNAs

**Xusheng Zhang, Wei Zhang, Yishu Gong, Liangliang Yang, Jianyu Zhang, Zhengyu Chen, Sihong He**
`xushengz@psu.edu, wei_zhang@g.harvard.edu, yig235@harvard.edu,`
`liangliang.yang@wsu.edu, zjianyu@umich.edu, chenzhengyu@zju.edu.cn,`
`sihong.he@uconn.edu`

## Abstract

Recently, multi-agent reinforcement learning (MARL) has shown its ability in solving sequential decision-making problems in complicated multi-agent environments. However, uncertainties from observations and executions undermine its performance when MARL methods are deployed in real-world applications. While crucial for deployment, a systematic robustness evaluation for MARL algorithms is not present. In this work, we utilize Gaussian noise attacks (GNAs) to examine the robustness of a benchmark MARL algorithm: multi-agent deep deterministic policy gradient (MADDPG). To the best of our knowledge, our work is the first to investigate the robustness of MADDPG to GNAs to observation and execution information. Our experiments show that GNA has totally different patterns in observation-wise attacks and execution-wise attacks. Furthermore, there are counter-intuitive insights from the experimental results which could guide researchers in future MARL methods development.

## 1 Introduction

Multi-agent reinforcement learning (MARL) has attracted increasing attention in recent years for its capability of solving real-world sequential decision-making problems which involve the interactions of multiple agents in a shared environment, for example, in game playing, traffic management, and robotics (Mnih et al., 2015; Sallab et al., 2017; He et al., 2022). However, uncertainties from observations and executions may degrade the performance of MARL algorithms and may yield unpleasant results in real-world scenarios (Tessler et al., 2019; Zhang et al., 2021; Dou et al., 2022a;b). Different sources of uncertainties, including measurement errors, model errors, operation errors, etc., need to be considered before algorithm deployments. Thus, to ensure that the MARL algorithms are reliable, adaptable, trustworthy, and suitable in a wide range of real-world applications, it is essential to evaluate robustness of MARL algorithms before their deployment (Pang et al., 2021). However, there is a lack of systematic robustness evaluations for MARL algorithms.

Researchers commonly quantify the robustness of machine learning (ML) methods by testing the performance of an ML algorithm after addicting Gaussian noise (a statistical noise with a Gaussian distribution) into the input (Pezzementi et al., 2018; Rauber et al., 2017; Turchetta et al., 2020). This approach, known as **Gaussian noise attack (GNA)**, provides a universal baseline robustness evaluation for ML methods. Therefore, in this work, we systematically evaluate the robustness of a benchmark MARL algorithm (multi-agent deep deterministic policy gradient, i.e., MADDPG (Lowe et al., 2017)) using GNAs. To the best of our knowledge, our work is the first to investigate the robustness of the benchmark MARL algorithm to Gaussian noise attacks to observation and execution information. We find that GNA has totally different patterns in the results of observation-wise attacks and execution-wise attacks. MADDPG's performance also highly depends on the multi-agent environment settings: in complicated environments, MADDPG can even achieve better performance under GNA than without attacks. Other counter-intuitive experimental results also help researchers to better design robust MARL algorithms (Yu et al., 2021).

## 2 Methodology and Experiments

**MADDPG** is a benchmark MARL algorithm that works very well for both cooperative and competitive environments (Lowe et al., 2017). We train MADDPG policies for agents in **MPE** (`https:`

`//github.com/openai/multiagent-particle-envs`), a benchmark multi-agent environment. There are 8 multi-agent scenarios in MPE: Mutual communication (MC), Cooperative communication (CC), Cooperative navigation (CN), Physical deception (PD), Encrypted communication (EC), Keep-away (KA), Predator-prey (PP) and Complicated game (CG). Each scenario is a multi-agent game that requires multiple agents to collaborate or/and compete. We provide detailed descriptions of these scenarios in Appendix C. The hyper-parameters we use to train our policies are provided in Appendix C, table 1. After each step of execution, we collect the reward received by each agent. By the end of the testing, which lasts for 10000 steps, we use agents' mean reward as a metric of the performance of MADDPG. In particular, we set this metric under no noise as our baseline. In the procedure of **Robustness Evaluation**, we inject a series of *i.i.d.* Gaussian noise into either the observation information (input of the policy) or the execution (output of the policy) when testing the well-trained MADDPG policies in each scenario. We use $\mathcal{N}(\mu, \sigma)$ to denote a Gaussian noise with mean $\mu$ and standard deviation (std) $\sigma$. See Appendix D for details of these attacks. To comprehensively understand the effect of Gaussian noise attack to MADDPG algorithm, we respectively set $\mu = -3, -2, -1, 0.001, 0.05, 0.1, 0.25, 0.5, 1, 2, 3$ and $\sigma = 3, 2, 1, 0.5, 0.25, 0.1$.

**Robustness to observation-wise GNA:** Depending on scenarios, robustness of MADDPG to GNA are distinct. See Fig. 1 and Fig. 3 for illustration. In scenarios MC, CC, CN, KA, and CG, under observation-GNA, there is a major decline on the agents' mean reward compared to the baselines (the mean reward of MADDPG under no noise). Interestingly, such decline even occurs under $\mathcal{N}(0, 0.1)$, which is supposed to closely resemble the baseline. Also unexpectedly, instead of decline, scenarios PP, EC and PD witness improvement of the agents' mean reward under GNA with certain parameters. Due to page limits, more experimental results are in Appendix E.

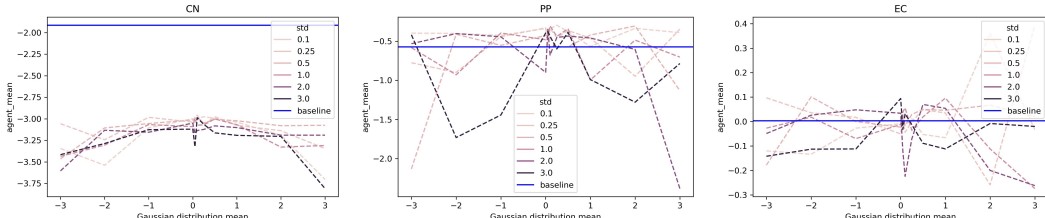

Figure 1: Agents' mean rewards under observation-wise GNA with different $\mu$ and $\sigma$.

**Robustness to execution-wise GNA:** In scenarios CN, CC, KA, the agents' mean reward is reluctant to the change of GNAs' mean values and MADDPG is robust to GNAs with small $\sigma$, i.e. the mean reward is below but close to that of the baseline. In scenarios PD, PP and CG, GNAs' $\mu$ has still limited influences on the agents' mean reward, but $\sigma$ plays a key role: although small $\sigma$ degrades the performance of agents as expected, large $\sigma$ even improves the agents' reward in PD compared to the baseline; moreover, we observe the inverse effect of $\sigma$ in PP and CG, that is, the agents' reward increases under GNAs with small $\sigma$ but decreases with large $\sigma$. See Fig. 2 and 4 for illustration. More experimental results are in Appendix E.

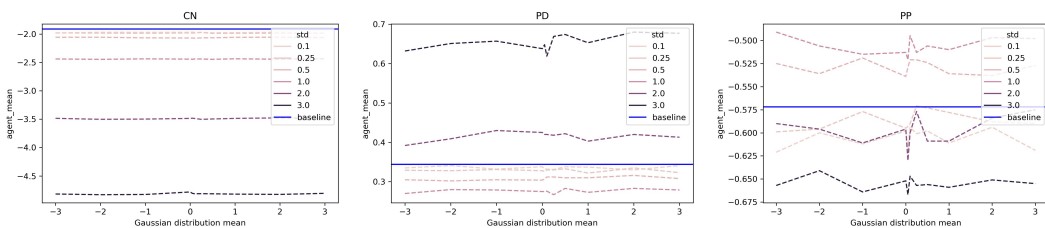

Figure 2: Agents' mean rewards under execution-wise GNA with different $\mu$ and $\sigma$.

**Conclusion and Future Work:** Considering that there is limited work regarding robustness evaluation for multi-agent reinforcement learning (MARL) methods, we conduct a systematic robustness analysis for a benchmark MARL algorithm with respective to observation and execution uncertainties, under Gaussian noise attacks. The counter-intuitive phenomena we find in the experiments could help researchers better design robust MARL algorithms. However, as real-world data noise sometimes follows a non-Gaussian distribution, it would be valuable to see other types of noise/attacks in robustness evaluation, which is considered as our future work.

URM STATEMENT

The authors acknowledge that last author of this work meets the URM criteria of ICLR 2023 Tiny Papers Track.

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

## A    PRELIMINARY: MARL AND MADDPG

Multi-Agent Reinforcement Learning (MARL) is a rapidly growing subfield of reinforcement learning (Zhao et al., 2022; Mei et al., 2023b), that focuses on developing algorithms and techniques for agents to learn how to collaborate and compete with each other (Zhou et al., 2022). MARL is relevant in many real-world applications such as traffic control (He et al., 2022), and game-playing (He et al., 2023a), where multiple agents need to coordinate their actions to achieve a common goal or compete with others. In general, we use a Markov Game to model a MARL problem (Littman, 1994; Owen, 2013). We use a tuple $G := (\mathcal{N}, S, \{A^i\}_{i \in \mathcal{N}}, \{r^i\}_{i \in \mathcal{N}}, p, \gamma)$ to denote a Markov Game, in which $S$ is the state space, $\mathcal{N}$ is a set of $N$ agents, $A^i$ is the action space of agent $i$. We define the joint actions space $A = A^1 \times \cdots \times A^N$. $\gamma \in [0, 1)$ is the discounting factor. The state transition function $p : S \times A \to \Delta(S)$ is a mapping from the current state and joint action to $\Delta(S)$, where $\Delta(S)$ represents the set of all probability distributions over the joint state space $S$. $r^i : S \times A \to \mathbb{R}$ is the reward function for $i$th agent. At time $t$, agent $i$ chooses its action $a_t^i$ according to a policy $\pi^i : S \to \Delta(A^i)$. We denote the joint policy as $\pi = (\pi^1, \cdots, \pi^N)$. For each agent $i$, its goal is to maximize its expected sum of discounted rewards, i.e. $J^i(s, \pi) = \mathbb{E}\left[\sum_{t=1}^{\infty} \gamma^{t-1} r_t^i(s_t, a_t) | s_1 = s, a_t \sim \pi(\cdot | s_t)\right]$.

Compare with single-agent RL problems, the MARL problem is more difficult to solve. Because from the perspective of each agent, they are facing a non-stationary environment. To solve this non-stationary issue, Lowe et al. (2017) proposed a multi-agent deep deterministic policy gradient (MADDPG) algorithm which adopts the framework of Centralized Training and Decentralized Execution (CTDE) and deep deterministic policy gradient (DDPG). DDPG is an Actor-Critic method and is based on the deterministic policy gradient algorithm (DPG). It uses a parameterized deterministic policy $\mu_\theta(s) : S \to A$ instead of a stochastic policy $\pi_\theta(a|s)$ in the objective function. The policy gradient has the following format:

$$\nabla_\theta J(\theta) = \mathbb{E}_{\tau \sim \mathcal{D}} \left[\nabla_a q^\mu(s, a) \nabla_\theta \mu_\theta(s)|_{a = \mu_\theta(s)}\right] \qquad (1)$$

CTDE framework enables MADDPG to extend actor-critic policy gradient methods to a multi-agent version where the critic can use extra information about the policies of other agents to ease training, while the actor is only augmented with local information. In an $N$-agent Markov game with a set of agent policies $\{\mu^1, \cdots, \mu^N\}$, the critic $Q_i(\mathbf{x}, a^1, ..., a^N)$ is a centralized action-value function that inputs all agents' action and some state information $\mathbf{x}$, outputs the Q-value for agent $i$. Here policies and critics are usually approximated by deep neural networks (DNNs). State information $\mathbf{x}$ could consist of the observations of all agents and also could include additional state information if available. The policy gradient is shown below:

$$\nabla_{\theta^i} J^i(\theta^i) = \mathbb{E}_{\tau \sim \mathcal{D}} \left[\nabla_{a^i} q_i^\mu(\mathbf{x}, a^1, \cdots, a^N) \nabla_{\theta^i} \mu^i(o^i)|_{a^i = \mu^i(o^i)}\right] \qquad (2)$$

## B    RELATED WORK

Since MARL recently achieved prominent performance in many decision-making applications (Dou et al., 2022c; Liu et al., 2022), researchers proposed many MARL methods, which can be generally divided into two categories: policy-based methods and value-based methods. Policy-based methods usually have an actor-critic framework, such as MADDPG (Lowe et al., 2017), COMA (Foerster & Assael, 2016) and MAAC (Iqbal & Sha, 2019). Value-based methods are usually used to solve collaborative games by factorizing the value function. For instance, VDN (Sunehag et al., 2018), QMIX (Rashid & Samvelyan, 2018), ReMIX (Mei et al., 2023a) can decompose the team value function into agent-wise value functions. Some researchers also adopt the idea of graph, such as Graph Neural Network (Li & Nabavi, 2023; Li et al., 2021; Xiao et al.; 2021; Chen et al., 2022) in developing MARL algorithms (Naderializadeh et al., 2020; Hu et al., 2021). However, without considering uncertainties from the environment, sensing, and execution, the performance of well-designed methods can be degraded when deployed in the real world (He et al., 2023b; 2020; Miao et al., 2021; Su et al., 2022; Hu et al., 2022). Adversarial training is empirically shown to improve agents' robustness to make the policies experience possible adversarial attacks (Chen et al., 2021). Pinto et al. (2017) formulate the robust RL problem as a minimax problem (Huang et al., 2023; Wu et al., 2023; Elmachtoub et al., 2023; Huang et al., 2021) then propose a method to train an agent in the presence of disturbance and obtain more robust policies. Zhang & Malkawi (2022) train the RL in real world environment with uncertainty and apply it in smart building control. There are

also some robust MARL methods proposed to defend state uncertainty (Han et al., 2022; He et al., 2023a), and model uncertainty (Zhang et al., 2020). However, there is a lack of systematic and universal robustness evaluation methods and protocols for MARL algorithms.

## C  MULTI-AGENT ENVIRONMENTS

In this section, we introduce the multi-agent environments (Lowe et al., 2017; Mordatch & Abbeel, 2017) we use in our experiments. By mixed game, we mean a game involving both cooperation and competition.

**Mutual Communication (MC)**   This is a cooperative game in which there are 2 agents, 3 landmarks of different colors. Each agent wants to get to their target landmark, which is known only by the other agent. Reward is collective, so agents have to learn to communicate the goal of the other agent, and navigate to their landmark. This is the same as the Cooperative Communication scenario where both agents are simultaneous speakers and listeners.

**Cooperative Communication (CC)**   This is a cooperative game same as Mutual Communication, except that one agent is the 'speaker' that does not move but observes goal of other agent, and the other agent is the listener who cannot speak, but must navigate to correct landmark.

**Cooperative Navigation (CN)**   This is a cooperative game in which there are 3 agents and 3 landmarks. Agents are rewarded based on how far any agent is from each landmark. Agents are penalized if they collide with other agents. So, agents have to learn to cover all the landmarks while avoiding collisions.

**Physical Deception (PD)**   This is a competitive game in which there are 1 adversary (red), 2 good agents (green), and 2 landmarks. All agents observe position of landmarks and other agents. One landmark is the 'target landmark' (colored green). Good agents rewarded based on how close one of them is to the target landmark, but negatively rewarded if the adversary is close to target landmark. Adversary is rewarded based on how close it is to the target, but it does not know which landmark is the target landmark. So good agents have to learn to 'split up' and cover all landmarks to deceive the adversary.

**Encrypted Communication (EC)**   This is a mixed game in where there are two good agents (alice and bob), one adversary (eve). Alice must sent a private message to bob over a public channel. Alice and bob are rewarded based on how well bob reconstructs the message, but negatively rewarded if eve can reconstruct the message. Alice and bob have a private key (randomly generated at beginning of each episode), which they must learn to use to encrypt the message.

**Keep-away (KA)**   This is a competitive game in where there are 1 agent, 1 adversary and 1 landmark. Agent is rewarded based on distance to landmark. Adversary is rewarded if it is close to the landmark, and if the agent is far from the landmark. So the adversary learns to push agent away from the landmark.

**Predator-prey (PP)**   This is a Predator-prey environment as well as a competitive game. 1 good agents (green) is faster and wants to avoid being hit by other 3 adversaries (red). Adversaries are slower and want to hit the good agent. Obstacles (large black circles) block the way.

**Complicated Game (CG)**   This is a mixed game similar to Predator-prey, except (1) there is food (small blue balls) that the good agents are rewarded for being near, (2) we now have 'forests' that hide agents inside from being seen from outside; (3) there is a 'leader adversary" that can see the agents at all times, and can communicate with the other adversaries to help coordinate the chase.

## D  EXECUTION-WISE ATTACKS AND OBSERVATION-WISE ATTACKS

In the training stage of MADDPG, agents receive rewards for their actions and use deep neural network to improve their policies. In each iteration, an agent performs these steps in order:

Table 1: Hyper-parameters

| Parameter | MADDPG |
|---|---|
| optimizer | Adam |
| learning rate | 0.01 |
| discount factor | 0.95 |
| replay buffer size | $10^6$ |
| number of hidden layers | 2 |
| activation function | Relu |
| number of hidden unites per layer | 64 |
| number of samples per minibatch | 1024 |
| target network update coefficient $\tau$ | 0.01 |
| iteration steps | 20 |
| episodes in training | 10k |
| time steps in one episode | 25 |

1. Select an action based on the agent's policy and execute it.

2. Observe the updated state resulting from the action and collect the corresponding reward.

3. Feed all of the information obtained so far, including the action and reward, into a deep learning algorithm, which updates the agent's policy.

During the robustness evaluation stage, in each group of experiment, depending on the step being evaluated, only one type of noise is applied, either to the action execution in step 1 or to the state observation in step 2 as follows.

1. During step 1, the agent's action selection is perturbed by adding random noise in the form of $\mathcal{N}(\mu, \sigma)$ to the action parameters before it is executed; in principle, the resulted action should deviate from the most desirable action over time.

2. In step 2, a noissy state observation is obtained by modifying the parameters of the originally observed state with $\mathcal{N}(\mu, \sigma)$; from the agent's perspective, this modified state appears as if it is a different state, potentially leading to sub-optimal evaluations and recalibrations of the agent's actions thereafter.

The former is known as *execution-wise attack*, and the latter is called *observation-wise attack*. The injected Gaussian noise $\mathcal{N}(\mu, \sigma)$ are always independent and identically distributed (*i.i.d.*).

## E   MORE EXPERIMENT RESULTS

**More discussions on observation-wise GNA:**  We study how the parameters $\mu$ and $\sigma$ affect the results of GNAs. In scenarios MC, CC, CN, KA, PP and CG, the parameters of observation-wise GNA influence the reward of agents in the following way: when $\mu$ is close to zero, the amount of degrade in agents' reward is not much related to $\mu$, but if the mean of noise is strongly biased (*i.e.* with large absolute value) to either positive or negative, the agents' reward further degrades, and it degrades symmetrically. For these scenarios, $\sigma$ also plays a role in the extent of the degradation, but there is no clear pattern (*e.g.*, monotonicity). In scenario EC, the effect of $\mu$ is no longer symmetric: the performance of agents has more variation under GNA with $\mu > 0$, and is less sensitive to negative-mean GNA. In scenario PD, GNA affects the performance of agents in an unexpected way: the agents perform even better under a strongly biased noise. See Fig. 3 and previous Fig. 1 for illustration.

**More discussions on execution-wise GNA:**  As for the execution-wise GNA, we also study the effects of different $\mu$ and $\sigma$. In scenario PP, when $\sigma$ of GNA is larger than 1, GNA with a larger $\sigma$ leads to a lower reward. However, when $\sigma < 1$, GNA with a smaller $\sigma$ leads to a lower reward. In the other 7 scenarios, GNA with a larger standard deviation has more power to undermine MADDPG, i.e. lower rewards.

On the other hand, $\mu$ affects agents' mean rewards on a case by case basis. MADDPG is not extremely sensitive to the means of GNAs in scenarios CC, CN, PD, PP and KA. Nevertheless, in

scenario MC, the larger $|\mu|$ is, the less reward the MADDPG algorithm can achieve. Remarkably in scenario EC with $\sigma \leq 0.25$, as $\mu$ increases to more positive, agents' reward increase. In scenario CG, the value of $\sigma$ affects the performance of MADDPG dominantly, and the mean of GNA makes a difference only if $\sigma$ is small. See Fig. 2 and 4.

**On the adversary's mean rewards under GNA:** For the mixed/competitive scenarios where adversary agents exist, we also check how the adversaries' rewards change under GNAs. See Fig. 5 for the adversary agents' mean rewards under observation-wise GNA and Fig. 6 under execution-wise GNA. First, the observation-wise GNA causes significant decrease in adversary agents' rewards in scenarios PD, EC, PP and CG, but in scenario KA, either decrease or increase could happen depending on the parameters of the GNA. Under execution-wise GNA, in scenario PD and PP, the adversary's rewards is mostly determined by $\sigma$. Particularly, if $\sigma$ is really small, the rewards are close to the baseline, if not better. In scenario EC, the adversary's rewards is decreased with larger $|\mu|$ and larger $\sigma$. In scenario KA, a large $\sigma$ drives adversary agents' reward sensitive to the change of $\mu$, contrasted to what is observed in scenario CG, but in both cases, the performance of MADDPG is worse than the baselines.

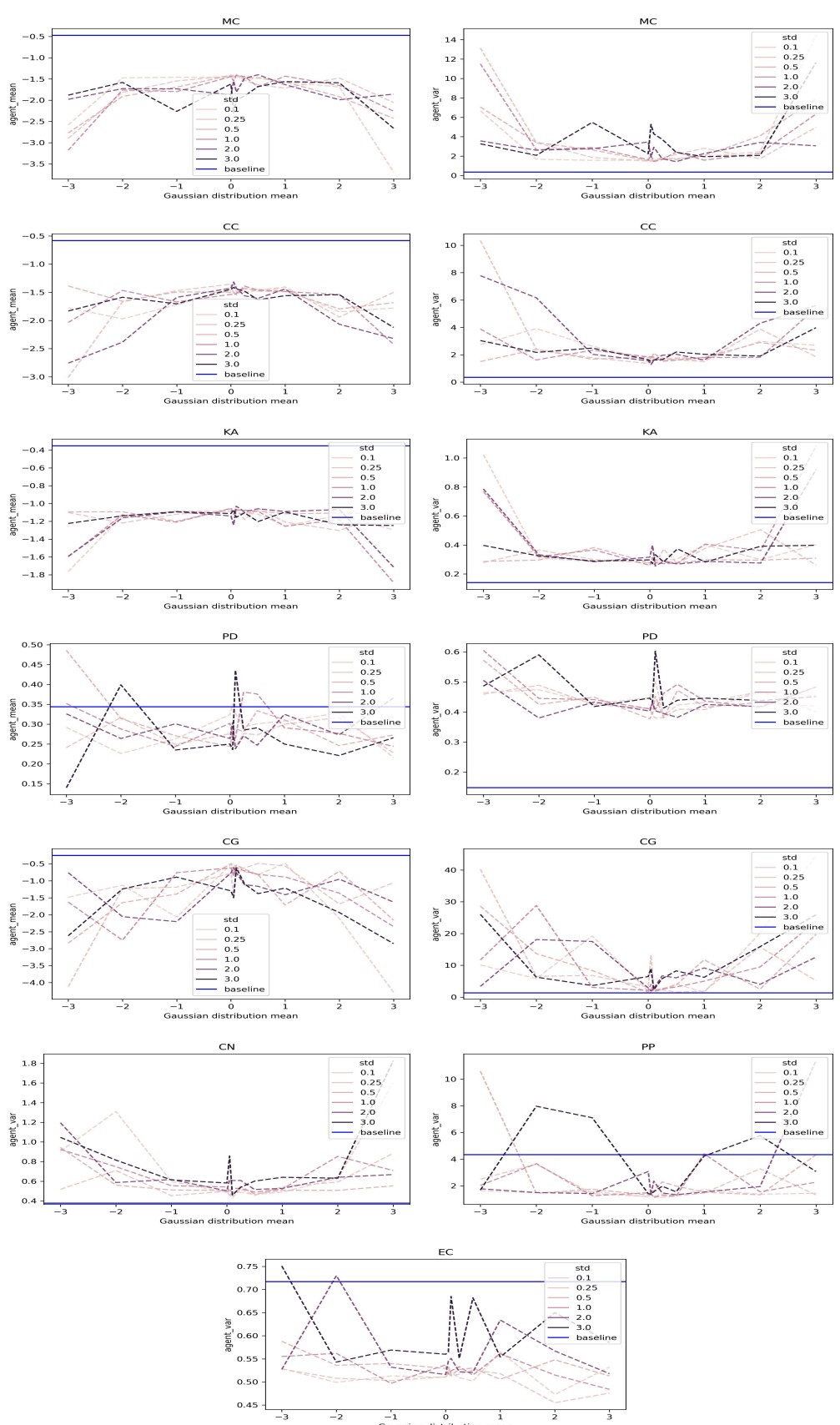

Figure 3: Agents' mean rewards and variance under observation-wise GNA with different $\mu$ and $\sigma$.

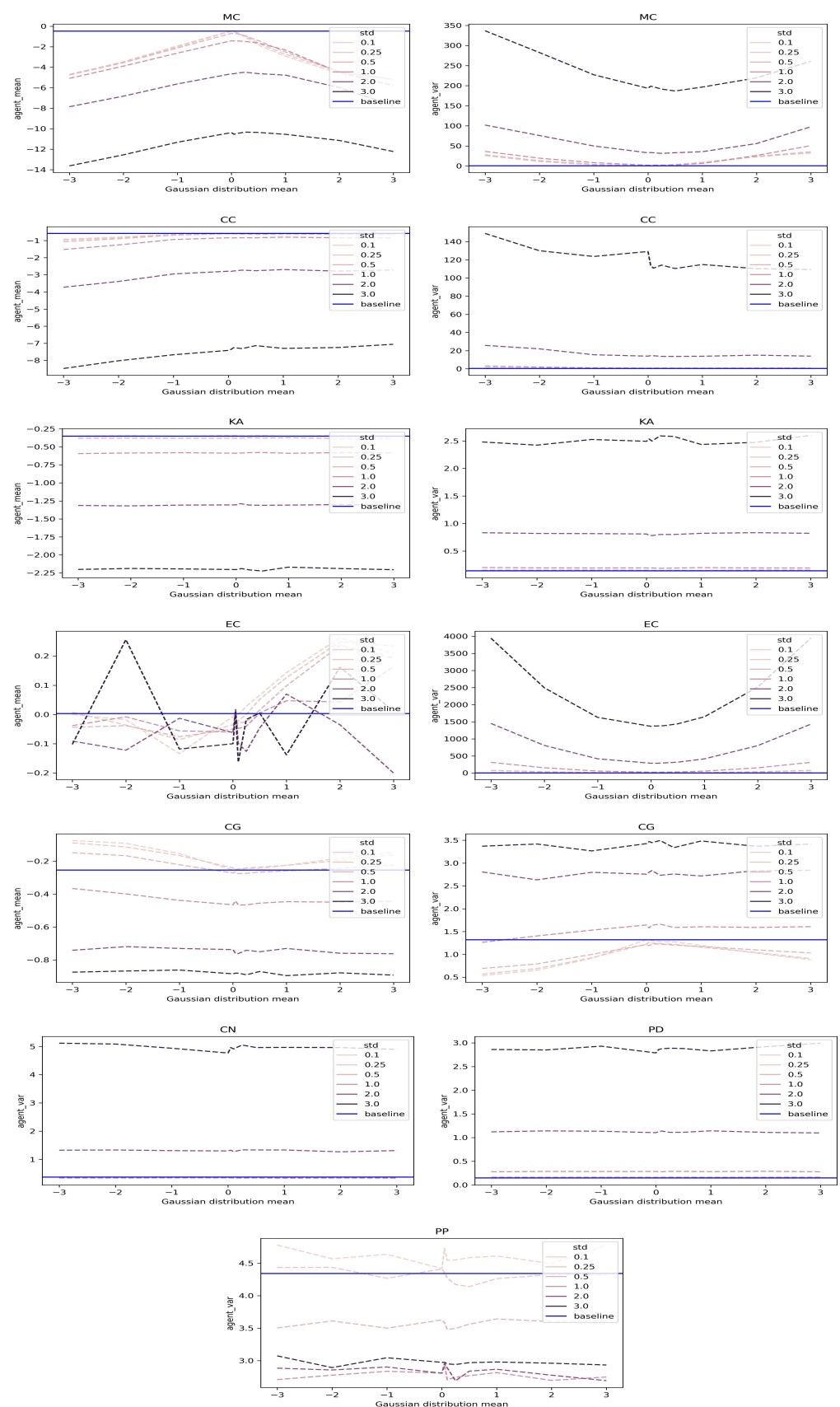

Figure 4: Agents' mean rewards and variance under execution-wise GNA with different $\mu$ and $\sigma$.

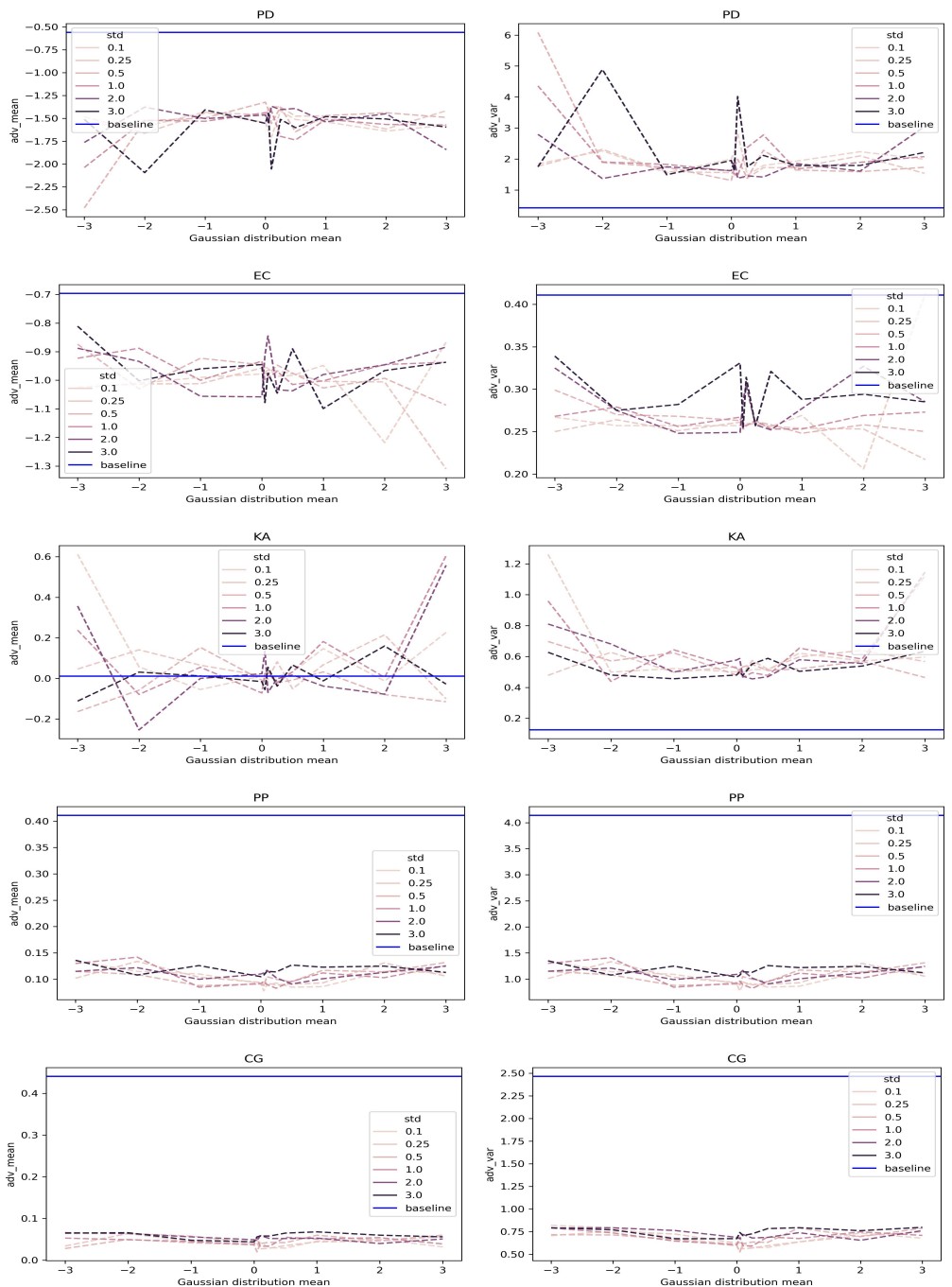

Figure 5: Adversary agents' mean rewards and variance under observation-wise GNA with different $\mu$ and $\sigma$.

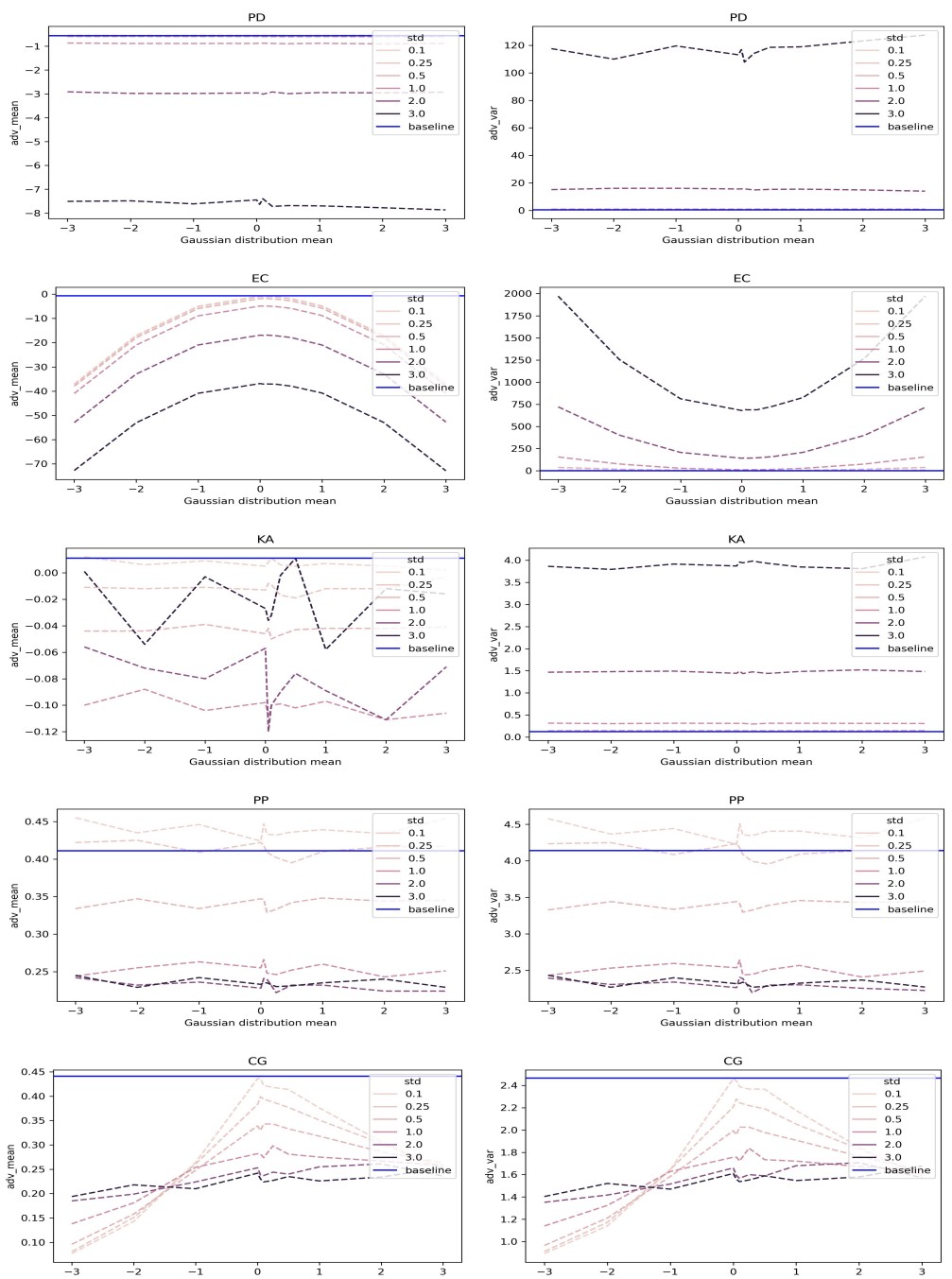

Figure 6: Adversary agents' mean rewards and variance under execution-wise GNA with different $\mu$ and $\sigma$.

