# OpenReview forum: "Robustness Evaluation of Multi-Agent Reinforcement Learning Algorithms using GNAs"
_ICLR.cc/2023/TinyPapers — Submitted to Tiny Papers @ ICLR 2023_

### Official Review · Reviewer_wnyg · 2023-03-20

**Confidence:** 5

**Summary Of Contributions:**

This work investigates the robustness of MADDPG towards gaussian noise attacks, specifically noise injected in the observation and environments. The experiments were performed on a multi-agent benchmark environment and the results shows MADDPG reacts differently towards the gaussian noise depending on the type of noise injection and the type of environment. Sometimes, the noise injection resulted in better performance, and sometimes it degrades the MADDPG performance.

**Rating:**

Great Start (GS): a submission which meets some of the reviewing criteria but has room for improvement

**Strengths And Weaknesses:**

Strengths:

1. The paper is relatively well written and easily understandable in terms of clarity and results.
2. The claims made by the authors also reflects the results shown, which I truly appreciate, especially since no overall trends were observed.
3. Basic submission requirements were met.

Weakness:

1. Certain parts of the paper in terms of discussion of relevant literature and description of the experiments to aid reproducibility could be improved upon.
2. The title is slightly misleading, as it seems to be more general than the actual experiments performed.



**Suggested Changes:**

1. The title of the paper is slightly misleading as it suggests that multiple MARL algorithms were evaluated. I suggest the authors change the title to be more concise since the experiments were only performed on MADDPG

2. In terms of discussion of related works, I understand that the authors may be restricted in terms of space to cite and discuss the many related works in literature. As such, perhaps the authors could cite more literature survey type papers focusing on robustness, rather than citing 1 or 2 specific papers related to adversarial attacks.

3. From the paper, it wasn't very clear what execution-based noise means. I could understand how the observation noise perturbations were performed, but it wasn't clear how the execution perturbations were achieved. It would greatly improve the reproducibility of the paper if the authors could add an additional paragraph on it.

4. I'm also curious if the results shown in the Figures were results averaged over multiple runs. As RL is famous for its sensitivity towards random initializations, it would make the results more convincing if the results were averaged over multiple runs. If the results were indeed already averaged, it would be helpful to show the standard deviations of the rewards as well.

All the best!

---

> ### Author Response · Authors · 2023-05-30
> **Reply to reviewer wnyg comments**
>
> Dear reviewer,
>
> We appreciate your valuable comments and suggestion for our tiny paper: "Robustness Evaluation of Multi-Agent Reinforcement Learning Algorithms using GNAs". After careful discussion following your advice, we improved our manuscript as the following:
> 1. we modified the title of our manuscript to focus better
> 2. we included  in Appendix B "RELATED WORK" more literature including papers on robustness
> 3. we added the explanation in Appendix "D EXECUTION-WISE ATTACKS AND OBSERVATION-WISE ATTACKS" for a better and detailed description of execution-wise attacks as well as observation-wise attacks
> 4. we confirmed that our experiment results are indeed averaged over time; and we added variance of reward plotting respectively in Figure 3,4,5, and 6 of Appendix.
>
> Best regards

---

### Official Review · Reviewer_CX7k · 2023-03-24

**Confidence:** 4

**Summary Of Contributions:**

The work experimentally investigates the robustness of a benchmark multi-agent reinforcement learning algorithm and presents some observations under different noise levels.

**Rating:**

Needs Clarification (NC): a submission which does not meet the reviewing criteria and needs clarification for its described problem or solution

**Strengths And Weaknesses:**

I appreciate the authors’ aim of comprehensively studying the robustness of MARL under attacks. However, the current work is too preliminary to draw any meaningful conclusion. Experiments with different platforms, MARL algorithms, and attack methods should be conducted. The authors observe that a large $\sigma$ improves the agents’ reward in some environments. Can the authors come up with an explanation for this and conduct experiments accordingly to verify their hypothesis?

**Suggested Changes:**

Please see the comments above.

---

> ### Author Response · Authors · 2023-05-30
> **Reply to reviewer CX7k comments**
>
> We sincerely appreciate your valuable feedback and have taken it into careful consideration. The current version of our work is indeed preliminary and requires further investigation to draw meaningful conclusions.
>
> Firstly, we fully agree with your suggestion to conduct experiments using different platforms, MARL algorithms, and attack methods to enhance the robustness analysis of MARL systems. In our study, we have already performed experiments across multiple platforms, including Windows, Mac, and Ubuntu, to ensure a diverse evaluation.
>
> Moreover, in our ongoing study, we are actively working on expanding our experimental evaluation. We aim to incorporate a broader range of noise attacks and MARL algorithms to provide a more comprehensive understanding of MARL robustness. Additionally, we have made adjustments to the title, abstract, and introduction to accurately represent the current state of our work.
>
> Regarding the observation that a large $\sigma$ improves the agents' reward in certain environments, we have developed several promising hypotheses that may explain this phenomenon. To confirm these hypothesis, we will design experiments that systematically evaluate its impact on the agents' learning performance. However, due to space limitations in the current study, we will further explore this topic in future research.
>
> We thank you once again for your valuable input, which will undoubtedly contribute to the enhancement of our work.
>
> Sincerely.

---

### Meta-Review · Area_Chair_YutX · 2023-04-06

**Recommendation:** Invite to present
**Confidence:** 5

**Metareview:**

This work explore the robustness of MADDPG towards gaussian noise attacks for MARL. This work is interesting and novel. Also, the performance seems to be nice. The presentation is also nice. However, the authors should conduct extensive experiments. Also, the English expression should be improved to make it more clear.

**Summary:**

This work provides a GNA as a benchmark for the MARL.

**Comments And Feedback To The Authors:**

Please refer to the meta review.

**Reason For Not Giving A Higher Recommendation:**

N?A

**Reason For Not Giving A Lower Recommendation:**

This research is useful for investigating the robustness of MARL.

---

### Decision · Program_Chairs · 2023-04-10

Invite to archive